# Fostering resilience in conflict-affected schools: A randomized controlled trial of the 3C program's effects on Afro-Colombian adolescents

resilience; compassion; promotion; adolescents; conflict

**Corresponding author:**
Lina María González-Ballesteros;
Email: lgonzalezb@javeriana.edu.co

Lina María González-Ballesteros[1,2,3] (iD), Mariana Vásquez-Ponce[2],

Oscar Eduardo Gómez-Cárdenas[2], Camila Andrea Castellanos-Roncancio[2,3,4],

Carlos Gómez-Restrepo[2], Sofia Pérez-Lalinde[3],

Sebastian Fernández de Castro-González[3], Luisa Fernanda González-Ballesteros[5],

Liliana Angélica Ponguta[6] and Viviana Alejandra Rodríguez[1]

[1]PhD Program in Clinical Epidemiology, Department of Clinical Epidemiology and Biostatistics, Faculty of Medicine, Pontificia Universidad Javeriana, Bogotá, Colombia (PhD candidate); [2]Department of Psychiatry and Mental Health, Faculty of Medicine, Pontificia Universidad Javeriana, Bogotá, Colombia; [3]Mental Health and Resilience Research Seedbed, Department of Psychiatry and Mental Health, Faculty of Medicine, Pontificia Universidad Javeriana, Bogotá, Colombia; [4]Fundación Saldarriaga Concha; [5]Northwell, New Hyde Park, NY, USA; Department of Pediatrics, Lenox Hill Hospital, New York, NY, USA; and Donald and Barbara Zucker School of Medicine at Hofstra/Northwell, Hempstead, NY, USA and [6]Tecnológico de Monterrey, Centro de Primera Infancia, Ave. Eugenio Garza Sada Sur 2501, 64849, Monterrey, N.L., México

## Abstract

Afro-Colombian adolescents in Tumaco face high mental-health risks due to armed conflict and structural marginalization. We tested the short-term efficacy of the 3C program to strengthen resilience, compassion, and prosocial behavior and to reduce anxiety, depression, and PTSD. Mixed-methods cluster RCT with concurrent triangulation; multilevel mixed-effects models with multiple imputation; assessments at baseline, 6, and 9 months. Resilience increased by 13.14 points at 6 months (large effect, d = 0.89) and remained elevated at 9 months. Anxiety and PTSD screenings were lower in the intervention group across follow-ups. Compassion and prosocial behavior improved at 6 months but attenuated by 9 months. Depression screenings decreased at 6 months and rebounded at 9 months. Qualitative data aligned with these patterns (students reported sustained use of stress-management skills and peer support). 3C demonstrated short-term efficacy for resilience, anxiety, and PTSD but showed limited durability for compassion, prosociality, and depression without ongoing reinforcement. The pattern of effect attenuation—particularly the complete depression rebound—indicates that 3C provides a foundational component requiring integration with booster sessions to sustain socioemotional gains.

## Impact statement

The Conmigo, Contigo, Con Todo (3C) program strengthens resilience, compassion, and socioemotional skills among Afro-Colombian adolescents in conflict-affected schools in Tumaco, Colombia. Using a culturally adapted cognitive-behavioral approach, 3C empowers youth to address challenges such as violence, racism, and systemic marginalization, fostering safe school environments that promote mental well-being. This community-based model, provided by teachers, is scalable and adaptable for other vulnerable settings, from Colombia to global conflict zones. By aligning with the 2030 Sustainable Development Goals, particularly mental health targets, 3C offers a initial efficacy in reducing anxiety and trauma in underserved populations, though sustained impact requires ongoing reinforcement through booster sessions. These strategies support Colombia's national mental health policies and provide a foundational model for developing and optimizing school-based mental health interventions in adversity-affected communities worldwide.

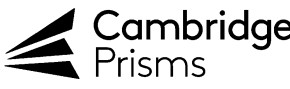



## Introduction

Tumaco, Colombia, a region impacted by persistent violence, armed conflict, and drug trafficking, continues to grapple with their repercussions even after the 2016 peace agreement with the FARC-EP (Fuerzas Armadas Revolucionarias de Colombia-Ejército del Pueblo). The protracted turmoil has particularly affected children and adolescents by compounding psychological burdens further

exacerbated by the COVID-19 pandemic, deepening social inequalities and heightening vulnerabilities within marginalized communities (Ramírez and Londoño, 2020; Figueroa and Valencia, 2021; Peltonen, 2024). Racism and systemic marginalization perpetuate discrimination and limit access to healthcare and education, disproportionately affecting Afro-Colombian adolescents in Tumaco and heightening mental-health risks (Bonilla-Escobar, 2021; Castro-Ramírez, 2021; Bonilla-Escobar et al., 2023).

In such challenging contexts, developing resilience, compassion, and prosocial behaviors **is essential to adolescents' ability to** navigate adversity and build healthy coping mechanisms. Resilience enables rapid recovery from difficulties, while compassion involves recognizing others' suffering and being motivated to alleviate it. Prosocial behaviors include empathy, helpfulness, and community engagement that strengthen social bonds and collective well-being (Penner et al., 1995; Connor and Davidson, 2003; López-Tello and Moreno-Coutiño, 2018). Moreover, these challenges underscore the urgent need for comprehensive, resilient, adaptive community-level strategies that address social vulnerabilities (displacement, violence, economic instability) and the broader environmental and systemic factors that perpetuate cycles of conflict and inequality (Rubio, 2005; Cerquera, 2020; Rosvold, 2023; Boston et al., 2024). Resolving these vulnerabilities is a long-term project; meanwhile, the 3C program offers a viable, ready-to-implement intervention.

Research shows that interventions targeting resilience and compassion can significantly improve adolescents' socioemotional competencies and their ability to thrive despite adversity, particularly in conflict-affected populations (Gilbert, 2014; Masten, 2015).

The *Conmigo, Contigo, Con Todo* (With Me, With You, With Everything - 3C) initiative has been implemented in Tumaco schools to promote resilience and compassion among adolescents facing armed conflict, forced displacement, and social inequalities. The 3C program leverages school-based activities to mitigate the effects of racism and marginalization (González-Ballesteros et al., 2021). Schools play a crucial role in the socioemotional development of adolescents and are thus pivotal venues for health promotion (Pulimeno et al., 2020; Santre, 2022). Implementing such programs positions schools as proactive agents in mental-health care, especially in areas that are conflict-affected, where traditional mental health services may be sparse. The 3C program activities include structured group discussions, role-playing exercises, and community service projects, designed to improve coping mechanisms and empathetic responses among students. These activities are expected to directly improve resilience and compassion, critical indicators of adolescents' psychological well-being (Ramírez and Londoño, 2020).

Given the high rates of PTSD, depression, and suicidal ideation among the region's youth, particularly in areas that are conflict-affected, it is crucial to integrate mental-health programs into community structures where they are most accessible. Schools serve as an optimal setting because students are consistently present, ensuring broader reach and impact. 3C builds resilience and compassion through interactive, supportive, school-based activities that mitigate psychological distress and promote mental-health awareness and empathy among adolescents. This cluster RCT evaluates 3C's efficacy in enhancing resilience and compassion among adolescents in Tumaco. The findings provide empirical evidence of benefit from targeted school-based mental-health interventions and may inform policy and educational strategies in similarly affected regions worldwide, contributing to global mental-health practice. Ultimately, the program supports adolescents' mental well-being and capacity to thrive in challenging environments.

## Methods

### Study design

This study was a cluster-randomized controlled trial with an embedded concurrent qualitative component, following Creswell and Plano Clark's (2017) mixed-methods framework and adhering to CONSORT-Cluster and GRAMMS reporting guidelines. Randomization occurred at the school level. Schools were assigned to either the 3C intervention or a wait-list control condition using a computer-generated sequence managed by an independent researcher. All eligible students within each school were assigned to their school's condition to preserve cluster integrity and ensure uniform exposure within clusters.

Quantitative and qualitative data were collected simultaneously at three time points: baseline, 6 months (endline), and 9 months (follow-up). The two data strands were analyzed independently before integration. Data triangulation was accomplished in the results phase by comparing quantitative outcomes (CD-RISC, ECOM, PSB scores) with corresponding qualitative themes (stress-management strategies, empathy reports, peer-support behaviors). Integration occurred through side-by-side comparison using joint displays in the Mixed Outcomes section, where quantitative statistical results were corroborated or contextualized by participant quotes and thematic patterns. Convergent findings strengthened interpretation, while divergent results were explored to understand intervention complexity.

### Participants

A total of 460 secondary school students in Tumaco, Colombia, were recruited (August 2023 to September 2024). Inclusion criteria were 12–18 years old and enrolled in one of the participating schools. Exclusion criteria included the presence of cognitive impairments that could hinder program engagement. Participant enrollment was conducted through school administrators who identified eligible students within targeted grade levels. Written informed consent was obtained from parents/legal guardians, and written assent was obtained from all adolescent participants prior to participation. Cognitive impairments were determined through teacher reports and school records rather than formal testing.

This study was reviewed and approved by the Institutional Research and Ethics Committee of the Faculty of Medicine at Pontificia Universidad Javeriana and the Hospital Universitario San Ignacio (Approval Act No. 17/2022).

### Sample-size planning

We used Kohn and Senyak's (2021) clinical-research calculators for the primary outcome (CD-RISC). Assuming a 4-point mean difference (SD = 8.81), $\alpha = 0.05$, and power = 0.80, the unadjusted requirement was ≈76 per arm (152 total). To account for clustering at the school/class levels, we applied the standard design-effect formula: $DE = 1 + (m - 1)\,\rho$, with ICC ($\rho$) = 0.03 Guarnizo-Guzmán et al., 2019 and the observed average cluster size (m). Thus, N_adjusted = N_unadjusted × DE, raising the minimum to ≈80 per arm (~160 total). Our achieved sample (n = 460 students) exceeded this target. Powering was based on student outcomes; teacher counts were not part of the power calculation.

### Randomization and blinding

Schools were randomly assigned to either the intervention or control group using a computer-generated sequence managed by

an independent researcher not involved in delivery or outcome assessment; allocation was concealed until assignment. Although the participants and facilitators could not be blinded due to the nature of the intervention, outcome assessors were blinded to group assignments.

## Intervention

The 3C program consisted of 12 bi-weekly, 90-min sessions, focusing on intrapersonal ("with me"), interpersonal ("with you"), and community ("with everyone") strategies that promoted long-term resilience, compassion, and prosociality (Figure 1). Sessions teachers delivered sessions during regular school hours as part of the curriculum. Each 90-min session incorporated cognitive-behavioral therapy principles, including cognitive restructuring, behavioral activation, and social skills training, along with elements of compassion-focused therapy. Sessions included structured group discussions, role-playing exercises, mindfulness activities, and community service projects. The program was developed by Fundación Saldarriaga Concha and was previously piloted with caregivers affected by armed conflict (González-Ballesteros et al., 2021) and then adapted for adolescents. No booster sessions were delivered between the 6- and 9-month assessments.

## Instruments

This study used instruments designed to assess resilience, compassion, and prosociality alongside secondary measures developed to screen for depression, anxiety, and PTSD.

### Resilience

**Resilience** was measured using the 25-item Connor–Davidson Resilience Scale (CD-RISC), which evaluates qualities considered essential for managing adversity, such as control, tenacity, personal competence, and tolerance for negative affect. Participants responded to 25 statements based on their reactions to similar experiences or anticipated behaviors in challenging situations. Each item is scored on a 5-point Likert scale, ranging from 0 (not at all true) to 4 (almost always true), with a total possible score of 0 to 100. Higher scores indicate greater resilience (Connor and Davidson, 2003). The CD-RISC has demonstrated robust reliability across diverse settings, including Colombia (Guarnizo-Guzmán et al., 2019).

### Compassion

The ECOM Compassion Scale measures motivation to alleviate suffering, affective reaction to suffering, and compassion toward animals; 17 items on a 1–5 Likert scale (never–always), higher scores = greater compassion; validated in Mexican populations (general, university, and Maya) (López-Tello and Moreno-Coutiño, 2018).

### Prosociality

The Prosocial Personality Battery (PSB; Penner et al., 1995) includes **56 items** (5-point Likert) covering empathy, ascription of responsibility, and helpfulness; higher scores indicate greater prosociality; cross-cultural applicability has been reported (Martí-Vilar et al., 2020).

### Secondary mental-health screening

Depression: Whooley Depression Screening (WDS; two yes/no items; any "yes" = positive screen) (Whooley et al., 1997; Ministerio de Salud, 2013). Anxiety: Hamilton Anxiety Rating Scale (HAM-A/ HARS), 14 items scored 0–4 (total 0–56). PTSD: PCL-C, 17 items scored 1–5 (total 17–85). Conflict exposure: single binary item

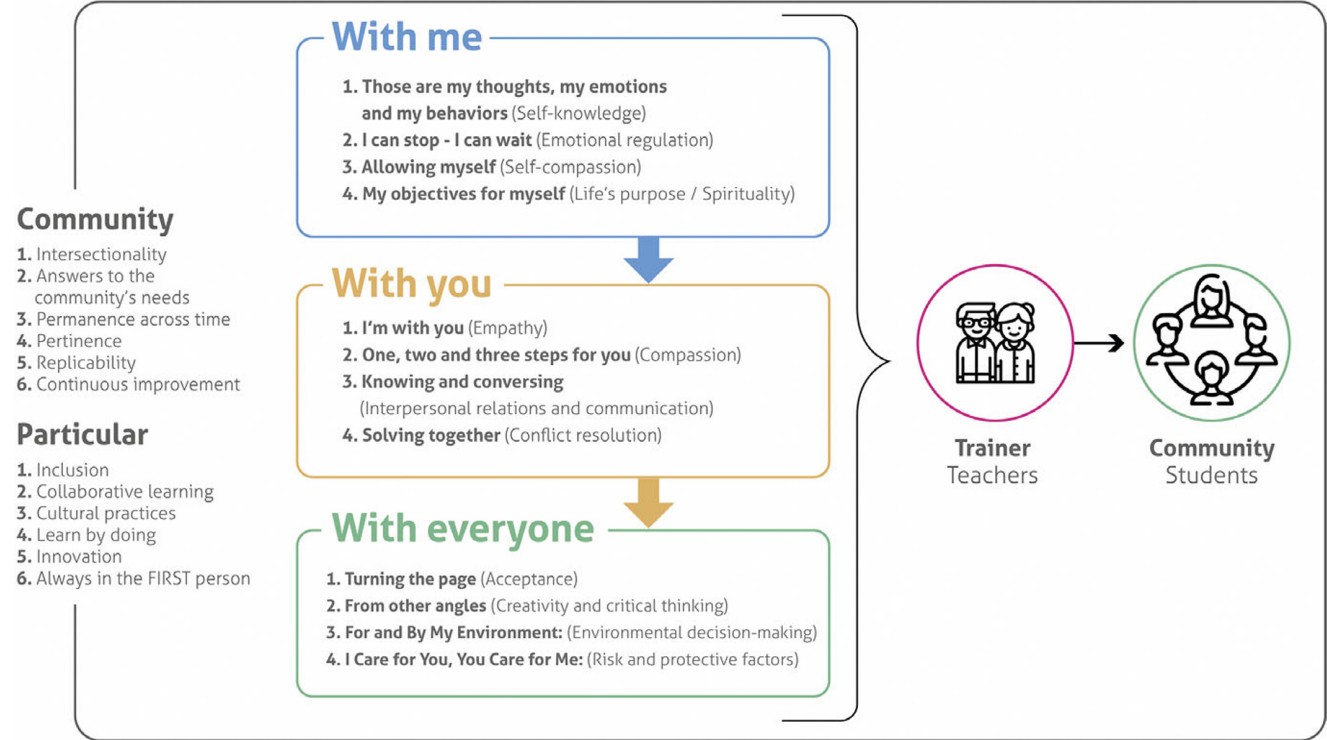

**Figure 1.** The 3C program aims to enhance resilience, compassion, and prosocial behavior in individuals and communities through a cognitive–behavioral framework that focuses on intrapersonal, interpersonal, and community strategies.

("Have you been a victim of armed conflict?" Yes/No) collected at baseline.

### Interpretation and meaningful change

The CD-RISC (0–100), ECOM, and PSB lack clinical cut-offs. Meaningful change was defined via effect sizes (Cohen's d: 0.2 small, 0.5 moderate, 0.8 large) and an MCID for CD-RISC of ~4–5 points (Connor and Davidson, 2003). For ECOM/PSB, changes were interpreted using effect sizes and triangulated with qualitative themes.

### Intervention fidelity and delivery.

Fidelity was maintained via a standardized manual. Facilitators (trained teachers) completed session checklists; kept field diaries; and collected end-of-session participant ratings (clarity/engagement). Focus groups were recorded and followed by feedback sessions. Facilitators completed a 2-day training led by psychologists, with competency assessed via role-plays (≥80% pass). Supervisors conducted random observations (~10% of sessions) across the five schools. Minor variations (e.g., brief scheduling shifts) were documented in diaries and considered in implementation notes.

### Data collection and analysis

Data were collected at baseline, 6 months (endline), and 9 months (follow-up post-intervention). Trained psychologists administered measures **in private classrooms** and fostered a supportive environment. The team (four psychologists) completed certification in human-subjects research and training in cultural sensitivity for Afro-Colombian youth. Data collection was supervised by the PI with regular QA meetings. Data were entered in REDCap, with 10% double-entry verification.

### Descriptive analyses

**Descriptive analyses** report frequencies/percentages (categorical) and central tendency/dispersion (continuous). For continuous outcomes (CD-RISC, ECOM, PSB, HAM-A, PCL-C), we fitted mixed-effects linear models with fixed effects for group (intervention vs. control), time (baseline, 6, 9 months), and the group×time interaction; random intercepts were specified for school and class (classes nested within schools). For the binary WDS, we used mixed-effects logistic regression. Missing data (~**5% at 9 months**) were addressed using multiple imputation by chained equations (MICE; m = 23): chained linear models for continuous outcomes and chained logistic models for WDS. Estimates from imputed datasets were combined using Rubin's rules in Stata 18 (StataCorp, 2023). We created a variable catalog for all models (Supplementary Table S2). As sensitivity checks, we (a) added baseline HAM-A and PCL-C as covariates in the CD-RISC model; (b) repeated analyses via complete-case models; comparisons are reported for the primary outcome (CD-RISC) (Supplementary Tables S3–S5).

### Qualitative strand

Nine focus groups (6–8 participants) were conducted at baseline, endline, and follow-up in private school settings by trained facilitators experienced with Afro-Colombian youth in conflict contexts. Sessions (60–90 min) were audio-recorded with consent using a semi-structured guide (resilience, compassion, socioemotional regulation). Transcripts underwent reflexive thematic analysis (Braun and Clarke, 2006). Two researchers independently coded, iteratively refined a codebook, and developed themes via constant comparison. Agreement was monitored via percentage agreement (>85%) with discussion to consensus. Trustworthiness was enhanced through member checking, peer debriefing, and an audit trail. Thematic saturation was estimated per Guest et al. (2020) as the proportion of new themes over total coded excerpts.

## Results

The study's randomization and follow-up process involved 460 participants, with 62.2% (n = 286) in the intervention group and 37.8% (n = 174) in the control group. Due to school changes or withdrawals from the education system, 18 students from the intervention group (6.3%) and five from the control group (2.9%) withdrew by the 9-month follow-up (Figure 2). Sociodemographic data are provided for the entire sample (n = 460), as are the scores obtained from each instrument at baseline and 6 months postintervention. Due to participant withdrawals, data from the 9-month follow-up are presented for the remaining 437 participants.

This diagram outlines the flow of participants through each phase of the trial, including eligibility assessment, randomization, intervention allocation, and follow-up. All participants who entered the study received the intervention or were placed on a waitlist by randomized allocation. All subjects in both groups participated in follow-up assessments at 6 months (100% retention). At 9 months, there was 5% attrition (total lost to follow-up n = 23; intervention n = 18 due to changed school or withdrawal, control n = 5 due to changed school or withdrawal). Attrition was influenced by social and political factors in Tumaco, Colombia.

### Baseline characteristics

The sample consisted of 244 (53%) females and 216 (47%) males, with an average age of 14.9 years (SD = 2.27). The intervention group contained slightly more males than females, and the control group was predominantly females. The gender distribution differed significantly between groups: the control group included 71.3% female participants (124/174) versus 42.0% female (120/286) in the intervention group ($\chi^2(1)$ = 36.14, $p < 0.001$). Because gender may correlate with several outcomes, all models adjust for gender, and we additionally assessed potential effect modification by gender on the main outcome of resilience. Most participants reported having a low socioeconomic background (Level 1)[1] and residing in urban areas. Exposure to armed conflict was higher in the control group (51.7%) than in the intervention group (32.5%). This baseline imbalance in conflict exposure represents an important study limitation that we addressed through effect modification testing. Approximately 54.6% of the victims are female, and 45.4% are male. Approximately 54.6% of the victims are female, and 45.4% are male. Nearly all participants identified as Afro-Colombian and were insured by the Colombian health system with social security (Table 1).

### Primary outcomes: Quantitative results

Measurement results are summarized in Figure 3 and Supplementary Table S1.

---

[1]The classification of participants as having a low socioeconomic background (Level 1) is based on the SISBEN system, the System for Identifying Potential Beneficiaries of Social Programs in Colombia. This system classifies individuals according to their living conditions and income levels. SISBEN 1 indicates explicitly the highest level of vulnerability (https://www.sisben.gov.co/paginas/que-es-sisben.html#:~:text=El%20Sisb%C3%A9n%20es%20el%20Sistema,a%20quienes%20 m%C3%A1s%20lo%20necesitan).

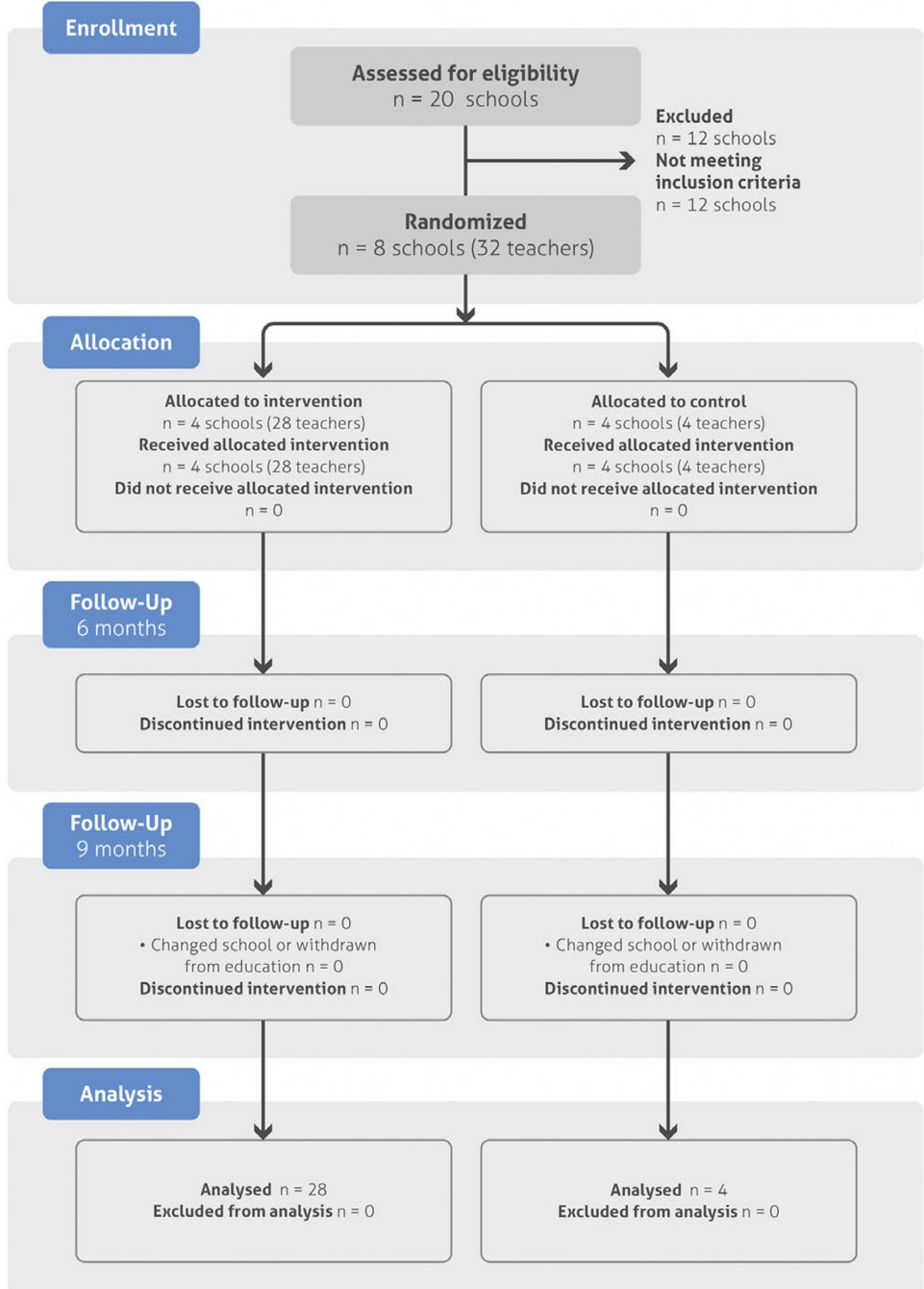

**Figure 2.** CONSORT flowchart.

*Resilience*

At each time point, the mean resilience scores (CD-RISC) for the control group were 65 points. This score was higher at baseline than that of the intervention group (61.88, SD = 15.91). At Time 2 (6 months postintervention), the intervention group's mean score increased to 74.91 (SD = 10.61). It then decreased to 66.36 (SD = 11.05) at Time 3 (9-month follow-up). Accounting for the interaction between time and the intervention assignment, a mixed-effects linear regression model was performed following multiple imputations of missing 9-month data to account for the 23 participants who withdrew from the study. Results indicated that the 3C program had a statistically significant effect on the participants' resilience. After adjusting for age, sex, and conflict exposure, the intervention group's mean score at 6 months was higher than the control group's (74.84 vs. 65.30; Table 2). Similarly, at Time 3, the intervention group's mean CD-RISC score was higher than that of the control group (66.30 and 65.21, respectively). These results reflect a mean increase of 13.14 (95% CI 10.27–16.00) in the intervention group's CD-RISC score at 6 months postintervention and a mean increase of 4.69 (95% CI 1.78–7.59) at 9 months postintervention. The 13.14-point increase at 6 months exceeded the minimal clinically important difference (MCID) of 4–5 points, with a large effect size (Cohen's d = 0.89, $p < 0.001$). The 4.69-point increase at 9 months met the MCID, with a moderate effect size (d = 0.43, $p = 0.002$), indicating clinically meaningful improvements.

**Table 1.** Sociodemographic characteristics by group and total

| Variable | Control (N = 174) | Intervention (N = 286) | Total (N = 460) |
|---|---|---|---|
| Educational institution | | | |
| IE 1 | 0 (0.0%) | 102 (35.7%) | 102 (22.2%) |
| IE 2 | 0 (0.0%) | 84 (29.4%) | 84 (18.3%) |
| IE 3 | 0 (0.0%) | 71 (24.8%) | 71 (15.4%) |
| IE 4 | 73 (42.0%) | 0 (0.0%) | 73 (15.9%) |
| IE 5 | 0 (0.0%) | 29 (10.1%) | 29 (6.3%) |
| IE 6 | 6 (3.4%) | 0 (0.0%) | 6 (1.3%) |
| IE 7 | 14 (8.0%) | 0 (0.0%) | 14 (3.0%) |
| IE 8 | 81 (46.6%) | 0 (0.0%) | 81 (17.6%) |
| Age (years) | | | |
| Mean (SD) | 14.86 (1.43) | 14.92 (2.66) | 14.90 (2.27) |
| Sex | | | |
| Female | 124 (71.3%) | 120 (42.0%) | 244 (53.0%) |
| Male | 50 (28.7%) | 166 (58.0%) | 216 (47.0%) |
| Socioeconomic status | | | |
| Level 1 | 161 (93.6%) | 272 (95.4%) | 433 (94.7%) |
| Level 2 | 11 (6.4%) | 11 (3.9%) | 22 (4.8%) |
| Level 3 | 0 (0.0%) | 2 (0.7%) | 2 (0.4%) |
| Housing location | | | |
| Rural | 1 (0.6%) | 2 (0.7%) | 3 (0.7%) |
| Urban | 173 (99.4%) | 284 (99.3%) | 457 (99.3%) |
| Victim of natural disasters | | | |
| No | 168 (96.6%) | 258 (90.2%) | 426 (92.6%) |
| Yes | 6 (3.4%) | 28 (9.8%) | 34 (7.4%) |
| Victim of armed conflict | | | |
| No | 84 (48.3%) | 193 (67.5%) | 277 (60.2%) |
| Yes | 90 (51.7%) | 93 (32.5%) | 183 (39.8%) |
| Disabled condition | | | |
| No | 171 (98.3%) | 277 (96.9%) | 448 (97.4%) |
| Yes | 3 (1.7%) | 9 (3.1%) | 12 (2.6%) |
| Ethnicity | | | |
| Afro-Colombian | 171 (98.3%) | 275 (96.2%) | 446 (97.0%) |
| Indigenous | 0 (0.0%) | 3 (1.0%) | 3 (0.7%) |
| None | 3 (1.7%) | 8 (2.8%) | 11 (2.4%) |
| SGSSS affiliation | | | |
| Yes | 174 (100.0%) | 283 (99.0%) | 457 (99.3%) |
| No | 0 (0.0%) | 3 (1.0%) | 3 (0.7%) |
| SGSSS status | | | |
| Contributive | 9 (5.2%) | 7 (2.4%) | 16 (3.5%) |
| Special | 1 (0.6%) | 0 (0.0%) | 1 (0.2%) |
| Subsidized | 164 (94.3%) | 276 (96.5%) | 440 (95.7%) |
| Unaffiliated | 0 (0.0%) | 3 (1.0%) | 3 (0.7%) |

*Note*: Institución educativa (IE) is Colombia's administrative denomination for elementary through senior high school schools. Socioeconomic status is based on the national classification (SISBEN, 2025), and SGSSS refers to the Sistema General de Seguridad Social en Salud, the national insurance system through which all individuals in Colombia have access to health services. Female proportion differs between groups at baseline (71.3% control vs. 42.0% intervention; $\chi^2(1) = 36.14$, $p < 0.001$). All models adjust for gender; no evidence of group×time×gender interaction was observed for the main outcome (CD-RISC).

Table 2 presents the coefficients from a mixed-effects linear regression model assessing the impact of the intervention on CD-RISC scores over time. Coefficients represent estimated change in CD-RISC scores per unit change in predictor.

*Compassion*
Compassion scores (ECOM) in the intervention group increased from a baseline mean of 62.52 (SD = 12.52) to 67.47 (SD = 12.05) at 6 months, followed by a decline to 61.32 (SD = 13.70) at 9 months postintervention. In contrast, the control group's scores remained relatively stable, hovering 65 points at baseline and both postintervention assessments. The group×time interaction showed a significant increase in compassion at 6 months for the intervention group (mean difference 4.42 points, 95% CI 1.69–7.15); however, based on a mean change of −1.50 points (95% CI [−4.27, 1.26]) this effect had diminished by the 9-month follow-up. Additionally, the regression model revealed that male participants had significantly lower compassion scores compared to female participants ($\beta = -1.9$, SD = 0.83, $p = 0.022$, 95% CI [−3.53, −0.28]). The 4.42-point increase at 6 months indicated a moderate effect size (Cohen's d = 0.46, $p = 0.001$), supported by qualitative reports of enhanced empathy. The −1.50-point change at 9 months was not significant (d = −0.15, $p = 0.286$), consistent with qualitative themes of sustained but diminished compassionate interactions.

Table 3 presents the coefficients from a mixed-effects linear regression model assessing the impact of the intervention on compassion scores at both follow-ups. The coefficients represent the estimated change in resilience scores associated with each predictor.

*Prosocial behavior (PSB)*
At baseline, both the intervention and control groups demonstrated similar prosocial behavior (PSB) scores, with a mean of approximately 85. At 6 months postintervention, the intervention group's scores rose significantly to 95.17 (SD = 7.40) but declined below baseline 9 months postintervention 81.42 (SD 6.33). In contrast, the control group's scores showed minimal changes throughout the study period and remained consistent near the baseline level. The group×time interaction revealed significant increases in prosocial behavior at 6 months (12.71 points, 95% CI 10.91–14.50) that were not sustained at 9 months (−0.16 points, 95% CI −1.98 to 1.65). Concerning sex differences, male participants scored significantly lower in prosocial behavior than females by −1.09 points (95% CI [−2.12, −0.05]). The 12.71-point increase at 6 months reflected a large effect size (Cohen's d = 1.12, $p < 0.001$), supported by qualitative themes of enhanced peer support. The −0.16-point change at 9 months was negligible (d = −0.02, $p = 0.860$), consistent with sustained community engagement reported qualitatively.

Table 4 shows the coefficients from a mixed-effects linear regression model assessing the impact of the intervention on the Prosocial Personality Battery scores over the time points at 6 and 9 months. The coefficients represent the estimated change in resilience scores associated with each predictor.

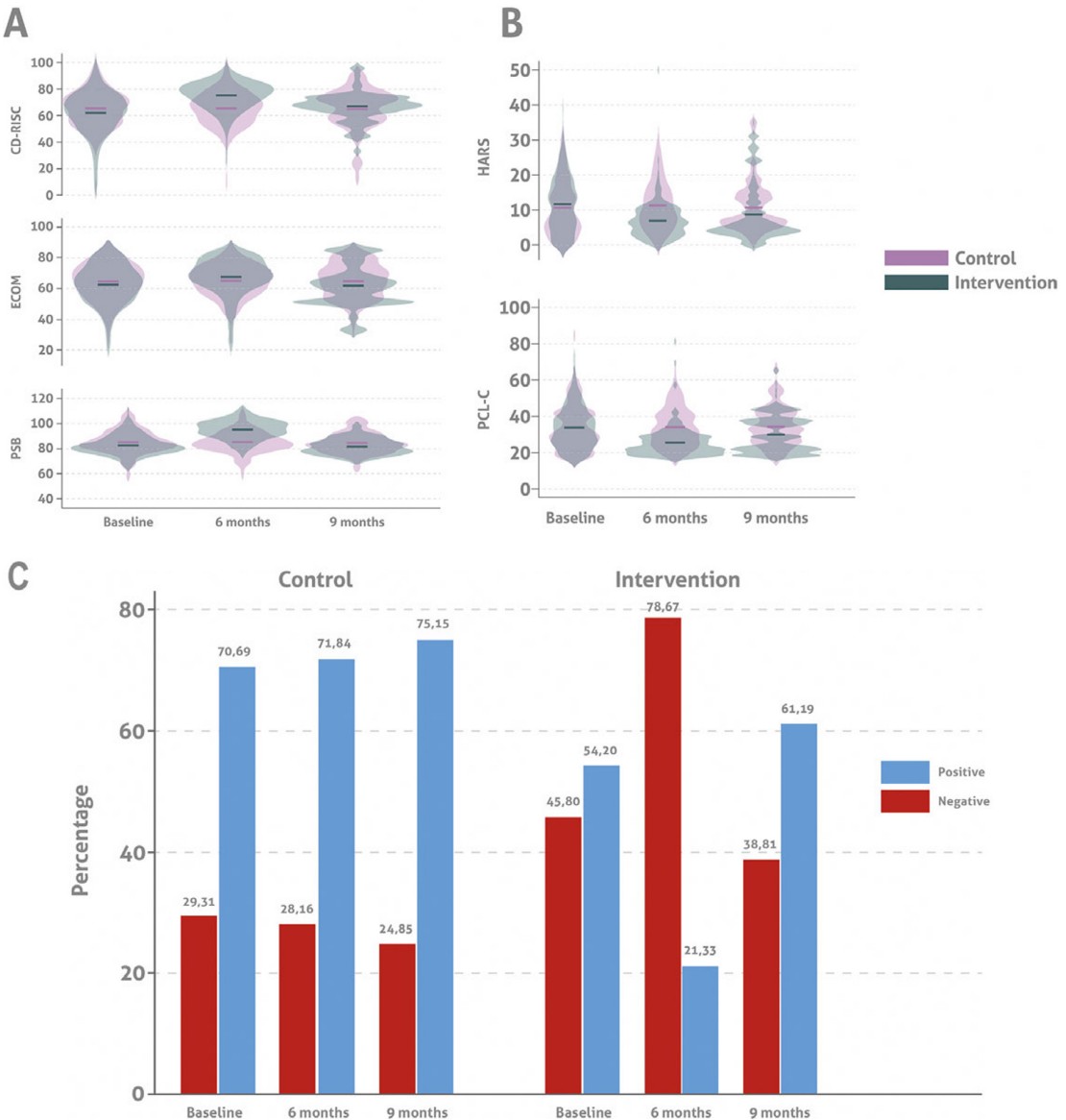

**Figure 3.** Measurement results at baseline and at 6 and 9 months postintervention. (A) Mean scores for resilience (CD-RISC), compassion (ECOM), and prosocial behavior (PSB) at baseline, 6, and 9 months postintervention by group. (B) Mean anxiety (HARS) and PTSD (PCL-C) screening scores at baseline, 6, and 9 months postintervention by group. (C) Percentage of positive and negative depression screenings (WDS) at baseline, 6, and 9 months postintervention by group.

### Mental health screenings

Anxiety trends. Initially, both the control and intervention groups displayed comparable levels of anxiety, with mean HARS scores of 10.76 and 11.50, respectively. At 6 months, anxiety decreased in the intervention group (mean 6.73) but increased slightly in controls (11.44). At follow-up, anxiety levels increased in both groups, with the intervention group showing a higher level of improvement, indicated by a follow-up mean score of 8.58 compared to the control group's 10.63.

Depression trends. Depression screenings at baseline showed that a larger portion of the intervention group was positively screened (54.2%) compared to the control group (70.7%). By endline, positive screenings in the intervention group dropped significantly to 21.3%, demonstrating a substantial reduction. However, follow-up data indicate a rebound in positive screenings, suggesting a partial resurgence of symptoms over time.

PTSD trends. Baseline PTSD screenings indicated a high rate of positive results in both groups. However, by the endline, the intervention group experienced a substantial decline in PTSD symptoms, with positive screenings reducing from 58.0% to 18.2%. This improvement was slightly reduced at follow-up, with an increase to 54.5% in positive screenings, yet this remained below the initial levels.

### Effect modification and sensitivity analyses

#### Gender effect modification analysis

To evaluate whether the baseline gender imbalance could bias the estimated intervention effects, we fitted a model that included a group×time×gender interaction term. Evidence for effect modification by gender was assessed using the joint significance of the three-way interaction contrasts at 6 and 9 months. We found no evidence that intervention effects varied by gender over time for CD-RISC. The magnitude and significance of the group×time effects were unchanged

**Table 2.** CD-RISC regression model results

|  | Coefficient | p value (0.05) | 95% CI | |
| --- | --- | --- | --- | --- |
| Group | | | | |
| Intervention | −3.6 | 0.005 | −6.13 | −1.06 |
| Time | | | | |
| 2 (6 months) | −0.11 | 0.925 | −2.37 | 2.15 |
| 3 (9 months) | −0.2 | 0.865 | −2.48 | 2.08 |
| Group–time interaction | | | | |
| Intervention–Time 2 | 13.14 | < 0.001 | 10.27 | 16 |
| Intervention–Time 3 | 4.69 | 0.002 | 1.78 | 7.59 |
| Age | 0.74 | < 0.001 | 0.35 | 1.13 |
| Sex | | | | |
| Male | 0.9 | 0.337 | −0.94 | 2.74 |
| Victim of armed conflict | 0.8 | 0.392 | −1.03 | 2.64 |
| Constant | 53.65 | < 0.001 | 47.54 | 59.77 |

**Table 4.** PSB regression model results

|  | Coefficient | p value | 95% CI | |
| --- | --- | --- | --- | --- |
| Group | | | | |
| Intervention | −2.17 | 0.005 | −3.66 | −0.67 |
| Time | | | | |
| 2 (6 months) | −0.17 | 0.811 | −1.59 | 1.24 |
| 3 (9 months) | −0.99 | 0.172 | −2.42 | 0.43 |
| Group–time interaction | | | | |
| Intervention–Time 2 | 12.71 | < 0.001 | 10.91 | 14.50 |
| Intervention–Time 3 | −0.16 | 0.860 | −1.98 | 1.65 |
| Age | 0.03 | 0.761 | −0.18 | 0.25 |
| Sex | | | | |
| Male | −1.09 | 0.040 | −2.12 | −0.05 |
| Victim of armed conflict | 0.40 | 0.449 | −0.64 | 1.44 |
| Constant | 84.79 | < 0.001 | 81.33 | 88.26 |

after adjustment for gender interaction testing ($p = 0.458$ for group×time×gender at 6 months; $p = 0.525$ at 9 months).

### Conflict exposure effect modification analysis

Given the baseline imbalance in conflict exposure, we tested whether intervention effects differed by victim of armed conflict (VCA) status using a model with group×time×VCA interactions. The primary group×time effects on resilience remained strong: at 6 months $β = 13.04$ (9.15–16.93, $p < 0.001$) and at 9 months $β = 5.81$ (95% CI 1.87–9.75, $p = 0.004$). The three-way interaction terms were $β = −0.03$ at 6 months ($p = 0.993$) and $β = −3.38$ at 9 months ($p = 0.266$), indicating no evidence of effect modification by VCA status. Sensitivity analyses stratifying by (and interacting with) VCA status suggest that the intervention's effect on resilience does not differ by exposure to armed conflict; any attenuation at 9 months among VCA participants was not statistically significant, supporting the robustness of our main findings.

**Table 3.** ECOM regression analysis results

|  | Coefficient | p value | 95% CI | |
| --- | --- | --- | --- | --- |
| Group | | | | |
| Intervention | −1.41 | 0.227 | −3.69 | 0.88 |
| Time | | | | |
| 2 (6 months) | 0.53 | 0.630 | −1.62 | 2.68 |
| 3 (9 months) | 0.31 | 0.782 | −1.86 | 2.48 |
| Group–time interaction | | | | |
| Intervention–Time 2 | 4.42 | 0.001 | 1.69 | 7.15 |
| Intervention–Time 3 | −1.5 | 0.286 | −4.27 | 1.26 |
| Age | 0.35 | 0.055 | −0.01 | 0.7 |
| Sex | | | | |
| Male | −1.9 | 0.022 | −3.53 | −0.28 |
| Victim of armed conflict | 0.52 | 0.527 | −1.09 | 2.13 |
| Constant | 59.71 | < 0.001 | 54.18 | 65.24 |

### Baseline mental health burden sensitivity analysis

To assess whether baseline mental health differences influenced intervention effects, we conducted a sensitivity analysis including baseline anxiety and PTSD symptoms as covariates in the CD-RISC model. Baseline anxiety was negatively associated with resilience ($β = −0.154$, SE = 0.070, $p = 0.028$), indicating that each 1-point higher baseline HARS score corresponded to approximately 0.15 points lower CD-RISC score. Baseline PTSD symptoms showed no significant association with resilience ($β = 0.068$, SE = 0.047, $p = 0.151$). Crucially, the intervention effects remained significant: group×time at 6 months $β = 11.94$ (SE = 1.94, $p < 0.001$) and at 9 months $β = 4.78$ (SE = 1.97, $p = 0.016$), consistent with the primary analysis. Complete results are provided in Supplementary Table S4.

Figure 3A,B summarize the results of the measurement instruments. In Figure 3A, the complete follow-up period shows that subjects in the control group had similar mean scores for resilience, compassion, and prosocial behavior. In contrast, participants who received the intervention displayed consistently higher mean scores for each measure at six months. This trend persisted at nine months for resilience but not for compassion and prosocial behavior, which both declined. Figure 3B presents the screening results for anxiety and PTSD, which were consistently lower in the intervention group at each follow-up point. These results suggest that the program's efforts to reduce mental health symptoms while strengthening socioemotional skills may have complementary effects.

Figure 3C shows a significant reduction in positive depression screenings at 6 months, followed by an increase at 9 months.

Figure 4 presents the interaction effects of the 3C program across three different metrics: Resilience (CD-RISC), Compassion (ECOM), and Prosociality (PSB) through all follow-up stages. **CD-RISC (Resilience): Control Group:** Starts at approximately 70, slightly rises to around 73 at the endline, and then decreases to 65 at follow-up. **Intervention Group:** Begins at around 65, climbs dramatically to nearly 78 at the endline, and drops to about 68 at follow-up. **ECOM (Compassion): Control Group:** Begins at 64.5, marginally increases to 65 at the endline, and falls back to 62 at follow-up. **Intervention Group:** Starts at approximately 62.5, rises to 67 at the endline, and then decreases to around 64 at follow-up. **PSB (Prosociality): Control Group:** Starts at 85, peaks at 95 at the endline, and returns to 85 at

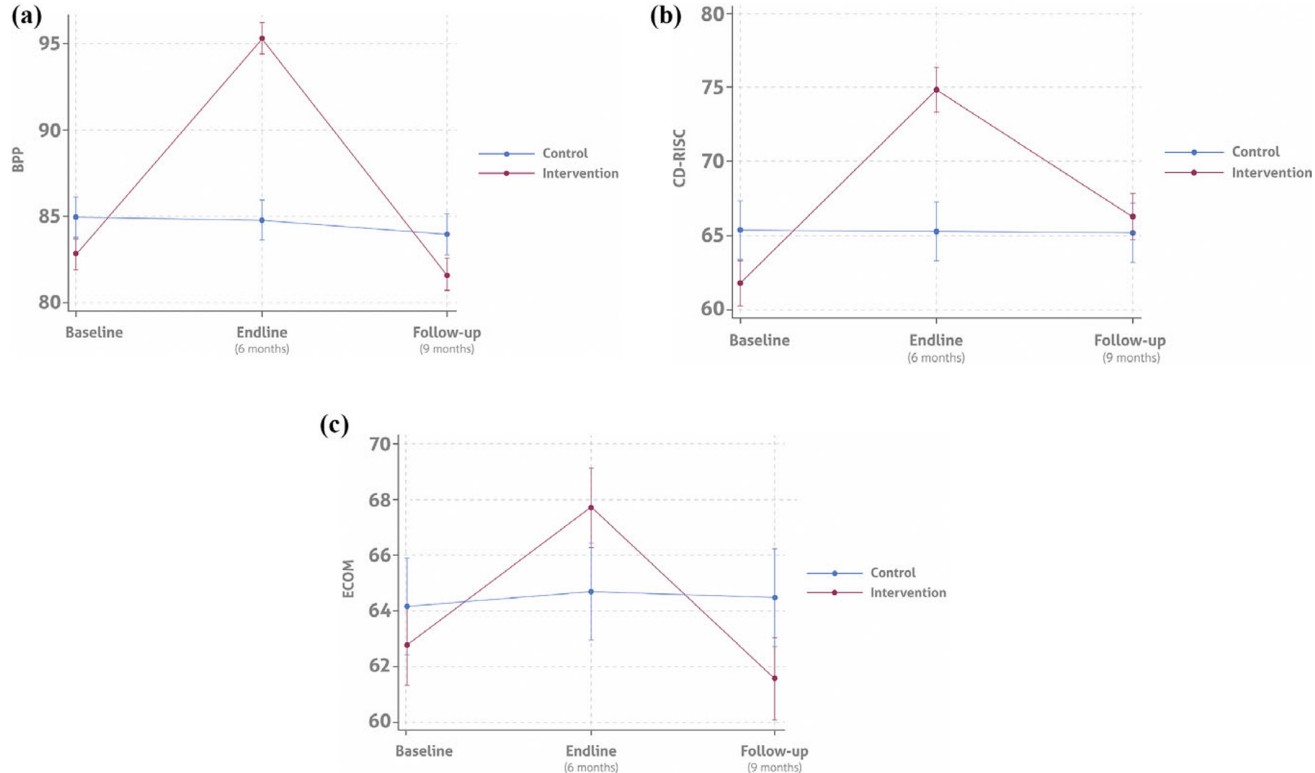

**Figure 4.** Interaction effect of the 3C program throughout the entire follow-up period.

follow-up. **Intervention Group:** Begins at 80, significantly increases to 95 at the endline, and drops to 82 at follow-up.

The analysis of the residuals versus fitted values for the CD-RISC, ECOM, and PSB models demonstrated a homoscedasticity pattern that indicated stable variability in the scores across the fitted levels. These findings support the robustness of the mixed-effects model in examining the association between intervention and resilience, compassion, and prosocial behavior among adolescents in conflict-affected regions (Figure 4).

### Primary outcomes: Qualitative results

The qualitative analysis explored the extent to which students perceived themselves as embodying the traits of the 3C framework—compassion, resilience, and socioemotional regulation—at baseline, endline (six months), and follow-up (nine months), assessing their thoughts and feelings about how they incorporated these traits and used relevant skills.

At baseline, students exhibited a nascent understanding of the 3C framework specifically, with thematic saturation achieved at 0.34, calculated following the method proposed by Guest et al. (2020). This coefficient represents the proportion of new themes identified relative to the total number of coded excerpts at that stage, indicating that while students demonstrated basic awareness of the 3C framework, their practical application was limited. Common themes included emotional reactivity and a general, albeit undeveloped, awareness of resilience and compassion. As one student reflected, "I would get angry quickly, and I didn't know how to calm down." This lack of self-regulation underscored their need for structured guidance in applying the 3C framework to their daily lives. Another student observed, "I try to help my friends, but sometimes I just don't know what to say." These reflections highlight students' early challenges in effectively implementing compassion, resilience, and socioemotional regulation in real-life situations. Their uncertainty and emotional reactivity underscore the importance of targeted interventions to help them develop and consistently apply these skills.

At endline, thematic saturation reached 0.63, indicating a broader range of responses and a greater engagement with resilience and compassion. Students' reflections highlighted emotional regulation as a developing skill, with frequent mentions of strategies for managing stress and emotional responses. As one student explained, "Now, when I feel angry, I remember to breathe and think before reacting." Reflective thinking and empathy also emerged as prominent themes, with students describing instances of compassionate interactions. One participant stated, "When my friend was having a tough time, I listened without judging and reminded them they're not alone." These insights indicate a shift from mere comprehension to the active application of 3C skills, particularly in the areas of empathy and interpersonal relationships.

At follow-up, thematic saturation peaked at 0.81, indicating a broad range of responses related to resilience, self-reflection, adaptive emotional regulation, and acceptance of change. Students frequently described strategies for managing emotions in various situations. As one student shared, "I still get nervous, but I use the breathing exercises, and it helps me stay calm." Compassion and empathy also emerged as recurring themes, with students reported consistently displaying empathy and understanding in peer interactions. One student described themselves as follows: "I try to support my friends without expecting anything in return; it just feels right to be there for each other. These responses suggest a continued engagement with the 3C framework, with students articulating their experiences of emotional regulation, self-reflection, and peer support in various contexts."

This progression in thematic saturation across measurements was accompanied by shifts in the depth and focus of student responses. Early discussions centered on basic awareness of emotional regulation, self-compassion, and empathy, often framed in terms of challenges or uncertainties. Over time, these themes became more detailed, with students describing specific strategies and reflections on their experiences. For instance, students initially expressed difficulty managing emotions, whereas at follow-up, they provided concrete examples of applying self-regulation techniques in real-life situations. As one student shared, "I still feel stronger in handling tough situations, like I can face things instead of feeling defeated. Similarly, discussions of compassion and empathy evolved from general acknowledgments of their importance to descriptions of active engagement in peer support.

### Mixed outcomes

Triangulation analysis revealed sustained impacts on resilience, anxiety, and PTSD, with partial maintenance of compassion and prosocial behavior. Quantitative follow-up measures indicated that while some initial gains decreased slightly over time, the intervention group demonstrated overall improvements compared to baseline levels. Regarding anxiety, for example, the intervention group's follow-up mean score was 8.58 (SD = 8.22), a substantial reduction from the baseline score of 11.50 (SD = 7.87). This trend aligns with qualitative reflections in which students described the lasting impact of the intervention in terms of their ability to manage stress. One student reported, "Even months later, I still use the breathing exercises when I get nervous, and they help me stay calm." These qualitative insights reinforce the quantitative findings, which showed that the students retained the coping strategies learned during the intervention.

Regarding resilience, the intervention group's follow-up CD-RISC scores were 66.36 (SD = 11.05), a significant improvement from the baseline levels. The mixed-model results indicate that these resilience gains were associated with the intervention ($\beta$ = 4.69, $p$ = 0.002). Qualitative data echoed this increase in the mean resilience score, with students continuing to report an increased sense of resilience and determination. At follow-up, one student shared the following: "I still feel more confident handling difficult things like I can find solutions instead of feeling defeated."

For depression quantitatively measured with the Whooley scale, follow-up screenings showed an increase in positive cases, from 21.3% at postintervention to 61.2%, which indicates a partial decline in the intervention's effectiveness over time. Nevertheless, qualitative feedback indicates that while some students experienced a relapse in depressive symptoms, many retained a more positive outlook than they had at baseline. One student said, "I still have hard days, but I know things can improve. I have hope now, which I didn't have before." This mixed finding suggests that while the intervention may initially reduce depressive symptoms, additional follow-up support might enhance its long-term impact on mental health.

Regarding compassion, the intervention group's ECOM scores increased from baseline (62.52, SD = 12.52) to a postintervention high of 67.47 (SD = 12.05) at 6 months but declined to 61.32 (SD = 13.70) at 9-month follow-up, falling slightly below baseline. Despite this quantitative decline, qualitative responses indicated that compassion remained present in the students' interpersonal interactions. Many described continued empathy and understanding in their peer relationships, with one student sharing the

following: "I try to listen to my friends when they need help and not judge them." This divergence may reflect measurement considerations including the ECOM scale's validation in Mexican rather than Afro-Colombian populations (López-Tello and Moreno-Coutiño, 2018), response shift bias where enhanced self-awareness led students to apply higher standards (Howard et al., 1979), and social desirability effects during active intervention. Their comment suggests that while compassion scores fluctuated slightly, students still exhibited compassionate behaviors, which suggests that the 3C program's teachings had a lasting effect on social dynamics.

Similarly, the intervention group's PSB scores increased from baseline (82.64, SD = 7.69) to a postintervention peak of 95.17 (SD = 7.40) at 6 months, then returned to approximately baseline levels at 9-month follow-up (81.42, SD = 6.33). Despite this pattern, qualitative data revealed sustained prosocial engagement. The students described a strong sense of community and how they actively supported each other in their daily interactions. One student explained this support by stating, "We're still looking out for each other, even if we don't have sessions anymore. It feels good to help." As with compassion, this pattern may reflect response shift and social desirability effects. This statement underscores the intervention's enduring influence on prosocial behaviors and suggests that the 3C program is effective at establishing a foundation for sustained peer support.

Ultimately, triangulating the quantitative and qualitative follow-up data suggests that while some immediate postintervention effects decreased over time, key improvements in anxiety, resilience, compassion, and prosocial behavior were largely sustained. The qualitative reflections confirm the quantitative trends and underscore the intervention's lasting impact on the development of essential socioemotional skills among students in high-risk settings.

## Discussion

This study provides evidence for the short-term efficacy of the 3C program—a mental health promotion strategy that targets resilience and socioemotional regulation—among adolescents in Tumaco, Colombia, a conflict-affected region. The pattern of results reveals differential sustainability across outcomes: resilience, anxiety, and PTSD showed sustained improvements at 9 months (3 of 6 primary outcomes), while compassion, prosociality, and depression demonstrated limited durability without ongoing reinforcement. These mixed findings demonstrate proof-of-concept status that the intervention requires further optimization and sustained support mechanisms rather than being policy-ready for widespread implementation.

### Mechanisms of differential sustainability

The pattern of results highlights which intervention components remained effective beyond the active implementation phase. Resilience gains persisted at nine months, suggesting that intrapersonal regulation skills—such as breathing exercises, cognitive reframing, and stress management—were internalized and self-reinforcing, allowing students to practice them independently. In contrast, compassion and prosociality declined, suggesting that interpersonal and community skills require continuous social reinforcement and practice opportunities, which diminished once the intervention concluded. This aligns with behavioral maintenance theory, which posits that skills requiring only individual practice sustain themselves better than those dependent on reciprocal social

interactions (Masten, 2015). Beyond skill-retention mechanisms, gender emerged as a significant moderator of intervention outcomes. The gender differential warrants further mechanistic exploration. Males demonstrated significantly lower compassion ($\beta = -1.9$, $p = 0.022$) and prosociality ($\beta = -1.09$, $p = 0.040$) across all time points. Our qualitative data suggest that these findings may reflect differences in engagement. In conflict-affected contexts, group activities focused on emotional expression may resonate less with masculine identity construction, particularly in cultural settings that value stoicism (Affleck et al., 2018). Future research should systematically analyze engagement patterns by gender to identify whether curriculum content, delivery methods, or peer dynamics contribute to these disparities.

### Alignment with national mental health frameworks

In Colombia, the current mental health framework—redefined by **Law 2,460 of 2025**, which updates Law 1,616 of 2013 and the *National Mental Health Policy*—positions resilience and emotional education as core principles for prevention, promotion, and comprehensive care in mental health (Ministerio de Salud, 2013, 2018; Congress of the Republic of Colombia, 2025). The 3C program aligns with these objectives by providing preliminary evidence supporting mental health promotion efforts that are key for validating and refining policy actions. Such evidence-based evaluations support the long-term effectiveness of mental health initiatives in high-risk youth populations. Similar studies on school-based interventions have highlighted the benefits of resilience and prosocial behavior programs. These programs create more supportive environments and equip students with essential coping mechanisms (Llistosella et al., 2023) particularly among vulnerable populations (United Nations, 2015). The school-based delivery model offers a promising approach adaptable across diverse cultural contexts, supporting the World Health Organization's emphasis on integrating mental health promotion into educational settings (WHO, 2021). The intervention's focus on resilience and compassion aligns with global priorities for building protective factors among youth exposed to adversity, providing proof-of-concept that requires further optimization and testing conflict-affected regions worldwide (Kieling et al., 2011).

This intervention model requires mechanisms for ongoing support and community ownership to achieve sustainable impact. Future iterations should embed maintenance strategies within the community. One key feature is first-person exercises, which facilitate participant internalization of skills and allow them to share these skills with peers, family, and community members, promoting broader socioemotional learning. This participatory approach aligns with Masten's (2015) research on resilience-building, which emphasizes active engagement and social reinforcement in fostering long-term behavior change. By creating opportunities for continued practice and integration, the 3C framework can sustain its benefits beyond the initial intervention phase.

The cultural and contextual adaptability of the 3C program underscores its broader application potential. Although environmental factors in conflict zones undoubtedly shape the experience and efficacy of mental health programs, intentionality and participant engagement remain equally powerful (Haine-Schlagel et al., 2021; Nizkorodov and Matthew, 2021; Fadhil and Aziz, 2023). As expressed by the participants, the self-driven desire for improvement and stress management supports culturally adapted, evidence-based, and regionally relevant interventions. Such approaches, conducted by local facilitators and rigorously

measured, provide a strong foundation for effective mental health promotion strategies (Betancourt and Khan, 2008; Manshoor and John, 2023).

Despite the 3C program's demonstrated efficacy, policy adaptations are necessary to enhance long-term sustainability and integration within Colombia's National Mental Health Plan. A critical recommendation is to establish follow-up strategies to reinforce learned skills and counteract gradual decline in intervention effects. Structured booster sessions may prevent symptom relapse and are supported by findings from similar school-based interventions (Bundy et al., 2011; Barry et al., 2013). Embedding these strategies within educational and community settings could ensure continuity, strengthen program adoption, and enhance long-term resilience among participants. By institutionalizing these follow-up mechanisms, the 3C program can maximize its impact and contribute to a more robust, scalable model for mental health promotion in high-risk contexts. The observed pattern of effect attenuation at 9 months reflects important considerations about systemic integration of school-based interventions. While resilience gains were partially maintained, suggesting successful internalization of core coping strategies, the decline in compassion and prosocial behaviors indicates these outcomes may require more sustained community-level reinforcement and ongoing practice opportunities. This pattern underscores that achieving lasting impact in complex, vulnerable environments requires not only demonstrating initial efficacy but also ensuring the intervention becomes embedded within existing educational and community systems.

While the intervention demonstrated sustained improvements in resilience and anxiety, the diminished effects for compassion, prosociality, and depression at 9 months warrant consideration. Several factors may explain these patterns. First, adolescence is characterized by rapid developmental changes requiring ongoing reinforcement (Shah et al., 2023). A critical review and meta-analysis of adolescent life skills programs in LMIC by Singla et al. (2020) highlighted that program effectiveness depends on multiple interacting factors, including therapy sessions, training practices, and therapeutic content—factors that collectively influence durability beyond initial implementation. Second, the 9-month assessment coincided with end-of-year adversity in Tumaco—job losses from expired contracts, reduced harvests, and escalated violence—contextual stressors that may have overwhelmed initial gains (Miller and Rasmussen, 2010). Third, teacher-led delivery, while scalable, may lack intensity for complex outcomes like depression. Singla et al.'s (2020) meta-analysis found small-to-medium effect sizes (SMD = 0.305) for depression in teacher-delivered programs, with substantial heterogeneity reflecting supervision and intensity variations. Shelemy et al. (2020) similarly found smaller effect sizes for depression versus anxiety in teacher-delivered interventions. Universal prevention programs delivered by non-specialists are effective for building general protective factors but may be insufficient to address clinical-level symptoms without additional specialized support (Patel et al., 2007; Shah et al., 2023). Fourth, limited parental involvement may have constrained durability; Singla et al. (2020) identified parent–child interactions as the strongest predictor of effectiveness ($\beta = 0.557$), noting that "skills reflecting parent–child relations were the least endorsed life skills subgroup but had the most relative influence of all life skills on trial effectiveness." While the 3C program included parent sensitization meetings, more intensive parent–child communication skill-building components may have enhanced durability, particularly for socioemotional outcomes strongly influenced by family

dynamics. Finally, the absence of booster sessions likely contributed to decay (Bundy et al., 2011; Barry et al., 2013; Singla et al., 2020; Shah et al., 2023). The depression rebound from 21.3% to 61.2% highlights the need for sustained mental health support. Unlike resilience skills—which students may practice independently—addressing depressive symptoms may require more intensive, specialized interventions beyond universal prevention. This pattern suggests that while 3C effectively builds protective factors, it may need complementation with targeted clinical services for adolescents with elevated depressive symptoms, particularly during periods of heightened environmental stress.

However, the depression rebound warrants careful interpretation beyond attributing it solely to intervention failure or contextual stressors; three measurement-related considerations may account for part of the observed pattern. First, the Whooley Depression Screening (WDS) is a two-item instrument designed for case-finding rather than longitudinal tracking, demonstrating high sensitivity to transient mood states (Whooley et al., 1997). The 9-month assessment timing at year-end—coinciding with documented stressors (job losses, harvest failures, violence escalation)—could have elevated positive screens through temporary mood reactivity without reflecting sustained depressive disorder. Second, the intervention explicitly taught emotional awareness as a core component, potentially enhancing students' capacity to recognize and accurately report depressive symptoms—representing improved mental health literacy rather than symptom worsening, a recognized phenomenon in psychological interventions (Kazdin, 2007). Third, the substantial baseline imbalance in depression screening (70.7% control versus 54.2% intervention) introduces regression to the mean: when groups differ at baseline, subsequent measurements naturally tend toward underlying mean values independent of intervention effects. The intervention group's "rebound" from 21.3% to 61.2% may partially reflect statistical regression toward their baseline (54.2%) rather than complete intervention failure. Standard covariate adjustment cannot fully eliminate confounding from such substantial baseline imbalances.

The divergence between quantitative screening results and qualitative findings supports this alternative interpretation. Students' sustained reports of hope, continued coping strategy use, and maintained optimism—even while screening positive on the WDS—suggest that the intervention helped establish foundational coping skills that persisted beyond active implementation. As one student stated, "I still have hard days, but I know things can improve. I have hope now, which I didn't have before." This exemplifies acknowledgment of ongoing challenges (potentially triggering positive WDS responses) alongside improved outlook and coping capacity. To distinguish true symptom recurrence from measurement artifacts, enhanced recognition, or regression to the mean, future research should employ multiple validated depression instruments suitable for repeated assessment, include ecological momentary assessment to capture real-time functioning, and use stratified randomization to improve baseline balance.

Regarding scalability beyond Tumaco, our findings provide specific insights for adaptation. First, our 95% retention rate at both follow-up points—exceptional for conflict-affected populations—suggests that school-based integration during regular hours with teacher-facilitators creates sustainable engagement that may not be achievable through external providers or extracurricular programming. This retention likely reflects the program's cultural adaptation to Afro-Colombian contexts and its integration within existing school structures rather than imposing external systems. Second, our observation that males demonstrated significantly lower compassion ($\beta = -1.9$, $p = 0.022$) and prosociality ($\beta = -1.09$, $p = 0.040$) across all time points indicates that current curriculum activities may resonate differentially by gender. In conflict-affected contexts where masculine identity construction may value stoicism (Affleck et al., 2018), programs may require tailored activities that engage male adolescents through culturally-appropriate expressions of resilience and peer support rather than focusing primarily on emotional expression and compassionate behaviors. Our qualitative data suggest that male participants engaged less in group discussions centered on feelings, pointing to potential curriculum modifications. Third, our pattern of results —teacher delivery proving effective for anxiety and PTSD but showing complete depression rebound by 9 months—has direct implications for scaled implementation. While universal prevention programs like 3C effectively build general protective factors, addressing clinical-level depressive symptoms requires integration within stepped-care models that link students screening positive for depression to mental health specialists (Patel et al., 2007; Shelemy et al., 2020). Relying solely on teacher-delivered universal prevention appears insufficient for depression outcomes, particularly in settings with seasonal stressors. Fourth, our 9-month assessment timing revealed vulnerability to predictable environmental stressors in Tumaco (year-end job losses from expired contracts, harvest failures, violence escalation). This finding suggests that booster sessions should be strategically scheduled before anticipated high-stress periods rather than at arbitrary intervals. In contexts with seasonal economic cycles or predictable conflict intensification patterns, aligning reinforcement sessions with these periods may enhance sustainability (Miller and Rasmussen, 2010). Finally, while Singla et al.'s (2020) meta-analysis identified parent–child communication skills as the strongest predictor of intervention effectiveness ($\beta = 0.557$), our program included only basic parent sensitization meetings. The decline in compassion and prosociality scores at follow-up—despite sustained qualitative reports of peer support—suggests that family-level reinforcement may be critical for maintaining interpersonal and community-focused outcomes. Future iterations should incorporate more intensive parent–child interaction components. However, cultural adaptation remains essential (Shah et al., 2023), as effective programs in LMIC demonstrate sensitivity to local contexts while maintaining fidelity to evidence-based components such as stress management and interpersonal skills (Singla et al., 2020).

The intervention improved compassion and prosocial behaviors at 6 months, with students reporting increased empathy and peer support. As one student remarked, "We're still looking out for each other," even if we do not have sessions anymore.', affirming sustained prosocial influence in high-risk settings (Wright et al., 2010; Ungar, 2013; Welford and Kasim, 2015).

Based on our findings, the 3C program effectively enhances resilience, reduces anxiety and PTSD symptoms, builds socioemotional competencies among adolescents in conflict-affected areas. Quantitative and qualitative data triangulation demonstrate its impact and suggest that periodic reinforcement may support longer-term efficacy, particularly for compassion, prosocial behavior, and depression outcomes. Future research should examine the optimal frequency and duration of booster sessions, thereby contributing further to the resilience literature and supporting the long-term socioemotional development of adolescents in adverse environments.

While this study offers valuable insights, several limitations must be considered. The measurement tools for resilience,

compassion, and prosociality lack established cut-off points, which may affect the accuracy and interpretation of the results due to social desirability or recall biases. Additionally, the ECOM scale was validated in Mexican populations rather than Afro-Colombian adolescents, which may affect cultural appropriateness and interpretation of compassion measurements. To mitigate this concern, we triangulated quantitative ECOM results with qualitative focus group data that captured participants' own culturally-contextualized descriptions of compassionate behaviors, strengthening our interpretation of compassion-related findings. In this regard, the scores reported throughout the entire study period demonstrate a trend similar to that observed in previous implementations of the 3C program, which indicated similar improvements in resilience (González-Ballesteros et al., 2021; Hein et al., 2024).

Attrition at 9 months (5.0%) may introduce selection bias if dropouts differed systematically from completers. Baseline imbalances in gender distribution (71.3% female in control vs. 42.0% in intervention, $\chi^2 = 36.14$, $p < 0.001$) and conflict exposure (51.7% vs. 32.5%) represent important study limitations. While all models adjusted for these covariates and interaction tests showed no evidence of effect modification, covariate adjustment cannot fully eliminate confounding when baseline imbalances are substantial, particularly in cluster-randomized designs where randomization occurs at school rather than individual level. Some proportion of observed intervention effects—particularly large effects at 6 months—may be partially confounded by these imbalances. Additionally, the control group's worse baseline mental health (70.7% vs. 54.2% depression-positive) raises the possibility that regression to the mean contributed to observed patterns. The 9-month follow-up may not assess long-term sustainability, and unmeasured confounding (e.g., social support, religiosity) or repeated instrument administration could influence results. Future research trials should employ stratified randomization to achieve baseline balance and include extended follow-up periods with larger, more diverse samples.

## Conclusion

The 3C school mental health promotion program demonstrated short-term efficacy in enhancing resilience and reducing anxiety/PTSD symptoms among adolescents in Tumaco, a conflict-affected region in Colombia. This study provides preliminary evidence that culturally adapted, school-based mental health interventions can produce meaningful improvements in selected outcomes —specifically resilience, anxiety, and PTSD—though effect attenuation for compassion, prosociality, and depression at 9 months indicates the need for ongoing reinforcement. This mixed pattern of results (sustained effects for 3 of 6 outcomes) suggests that 3C effectively builds foundational protective factors but requires integration with booster sessions and, for depression, complementary clinical services to achieve durable impact across all targeted domains. The partial maintenance of positive effects at follow-up underscore the intervention's potential as a foundational component and highlight the need to integrate similar programs into existing mental health frameworks, particularly in high-risk environments. The qualitative insights revealed that students internalized the skills learned through the program, illustrating the profound influence of culturally adapted community-based interventions. However, the attenuation of effects for compassion, prosociality, and depression at 9 months—particularly the rebound in depression screenings—underscores the critical need for booster sessions and ongoing

reinforcement to sustain socioemotional gains (Singla et al., 2020; Shah et al., 2023). Future research should examine the program's effectiveness (as distinct from the short-term efficacy demonstrated here) when implemented at scale, assess optimal booster session frequency and timing, and evaluate the long-term sustainability of the 3C program model across diverse contexts to meet the evolving mental health needs of adolescents worldwide. Ultimately, this study reinforces the critical role that structured mental health initiatives play in fostering resilience and emotional well-being in vulnerable populations.

**Open peer review.** To view the open peer review materials for this article, please visit http://doi.org/10.1017/gmh.2025.10119.

**Supplementary material.** The supplementary material for this article can be found at http://doi.org/10.1017/gmh.2025.10119.

**Data availability statement.** Data are available upon reasonable request to the corresponding author (lgonzalezb@javeriana.edu.co), subject to ethical approval and de-identification to protect participant privacy.

**Acknowledgements.** We would like to extend our heartfelt gratitude to the entire educational community of Tumaco (Nariño) for their invaluable support and participation in this study. Special thanks to Javier Andrés López for his dedication and commitment to implementing the 3C program and to Jeimmi Carvajal for her assistance in the qualitative data collection and analysis. Their contributions were instrumental in making this research possible. We express our heartfelt gratitude to Drs. López Tello and Moreno Cutiño for their invaluable contributions to our research. Their generosity in providing us with the ECOM instrument significantly enhanced the depth and breadth of our study. Their support was instrumental in facilitating our exploration of compassion within the school environments of Tumaco, Nariño. We thank Chelsea Larsson from Scribendi (www.scribendi.com) for editing a draft of this manuscript.

**Author contribution.** LMG-B led the study design, conducted the research, performed data analysis, and was primarily responsible for writing the manuscript. CG and VRR contributed to the methodology development and guided the methodological framework. CG and VRR also assisted in refining the methodology, conducting analyses, and reviewing the manuscript for critical revisions. OG supported quantitative data analysis and contributed to data collection. CC-R assisted the qualitative data analysis, collected data, and contributed insights to the thematic evaluation. MV-P contributed to manuscript writing and provided analytical support throughout the drafting process. SP-L and SFdC supported the analyses and assisted in writing the manuscript, thereby contributing to the overall quality and integrity of the research. LAP reviewed the manuscript for critical revisions.

**Financial support.** This research received funding from Fundación Saldarriaga Concha, which supported study design and data collection. LMG-B and CC-R are employed by Fundación Saldarriaga Concha. The funder had no role in data analysis, results interpretation, or manuscript preparation, which were conducted independently by the research team.

**Competing interests.** The authors declare none.

**AI disclosure statement.** During manuscript preparation, the authors used Claude AI for language editing, grammar checking, and formatting assistance. AI was not used for data analysis, results interpretation, or conceptual development. All scientific content, methodology, analysis, and conclusions remain entirely the responsibility of the authors. The final manuscript was reviewed and approved by all authors.

**Ethics statement.** This study was reviewed and approved by the Institutional Research and Ethics Committee of the Faculty of Medicine at Pontificia Universidad Javeriana and the Hospital Universitario San Ignacio, in an ordinary session held on September 22, 2022 (Approval Act No. 17/2022). Written informed consent was obtained from all participants and/or their legal

guardians. Assent was also obtained from all adolescent participants. All procedures followed the ethical standards of the Declaration of Helsinki and its later amendments.

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
