## [Reviewer Report]

This is a good manuscript presenting a well-executed, cluster-randomized controlled trial of the 3C program—a school-based psychosocial intervention targeting adolescents in a conflict-affected region of Colombia. The study addresses a critically important issue in global mental health: the promotion of resilience and emotional well-being among youth in humanitarian settings. The use of mixed methods and the cultural contextualization of the intervention are commendable.

Major Strengths

1. Relevance and Innovation:

The focus on Afro-Colombian adolescents and the use of a culturally grounded framework (3C: Conmigo, Contigo, Con Todo) make this a valuable contribution to the literature on adolescent mental health in conflict zones.

2. Robust Design:

The study design is rigorous, employing a cluster-randomized controlled trial with a mixed-methods approach and appropriate statistical modeling (including multilevel mixed-effects regression and imputation).

3. Integration of Qualitative Insights:

The qualitative findings, including progression in thematic saturation, provide depth and context to the quantitative outcomes and enrich the overall interpretation of program impact.

4. Policy and Practice Implications:

The manuscript effectively connects findings to Colombia’s national mental health policy framework and highlights the intervention’s scalability.

Suggestions for Improvement

1. Clarify Intervention Fidelity and Delivery:

Please provide more detail on how fidelity to the 3C program was monitored across schools and facilitators. Were there notable variations in delivery that might influence results?

2. Language and Editing:

Several grammatical issues and minor typographical errors were observed (e.g., fragmented definitions of resilience and compassion on page 3, redundant phrases in the results section). Handling of Attrition:

The manuscript notes participant dropout and use of multiple imputation. It would be helpful to include a brief sensitivity analysis or discussion of potential bias due to attrition.

3. Measurement Cutoffs and Interpretation:

The instruments used (e.g., CD-RISC, PSB, ECOM) do not have standard clinical cutoffs. Consider briefly discussing how meaningful change was determined, particularly for compassion and prosociality.

---

## [Reviewer Report]

Thanks for the opportunity to review this manuscript. The paper addresses a relevant and sensitive topic through a solid mixed-methods approach, evaluating a school-based intervention in a complex context. The focus on Afro-Colombian adolescents in conflict-affected areas is important, and the study brings useful insights for both academic and policy discussions. I organized my comments in bullet points by ascending page and line numbers.

Page 2, lines 49–53: The abstract mentions that the 3C program was effective. Please clarify what definition of “effectiveness” is being used. Does it refer only to internal efficacy in measured outcomes, or does it also imply long term collective or systemic impact, particularly given the references to sustainability?

Page 3, Introduction: The introduction would benefit from more explicit discussion of social determinants and context, such as racism, marginalization, or systemic exclusion, which affect Afro-Colombian adolescents in Tumaco and may influence both baseline vulnerability and intervention outcomes. In complex social systems, the context is an element that shapes both the behaviors being observed and the observer’s position. Acknowledging how systemic conditions, such as exclusion, inequality, or institutional neglect, co-construct the phenomena an intervention seeks to address is essential for understanding its dynamics, limits, and potential for sustainable impact.

Page 6, lines 19-26: The ECOM scale for compassion was validated in Mexico with university and Indigenous populations. However, it is not clear is that population is similar to the population of this study. Please discuss its cultural, age and linguistic appropriateness for Afro-Colombian adolescents in Tumaco. If local validation was not conducted or the population is different from the mexican population you are mentioning, this should be acknowledged as a limitation.

Page 7, lines 3-28: While the methods section shows the general modeling strategy, it should show more details regarding the analytical setup. I recommend including a supplementary table detailing all variables used in the regression models, including their names, categories or scales (binary, categorical, continuous), possible values, and any transformations applied. Additionally, the explanation of the multiple imputation process is somewhat limited. I recommend providing in a supp material more detail on the percentage of missing data per variable, the specific imputation methods used (predictive mean matching, logistic regression, etc), the number of imputations performed, etc. Regarding thematic analysis, it is unclear how many researchers were involved in coding, how saturation coefficients were calculated and interpreted, how disagreements were addressed, or whether there was triangulation across researchers to mitigate potential bias. Given that thematic interpretation can be shaped by the perspectives of those involved, I encourage the authors to clarify whether any reliability checks were conducted or if a process of consensus building was used. Also, I recommend to do a reflection on researcher positionality. In complex sociocultural settings such as this, the observer’s role can influence the framing and meaning of the themes that emerge.

Page 8, line 42: please explain in numbers how was predominantly females the control group. This could introduce a bias in the results?

Page 8, lines 43-45: There is an important imbalance in conflict exposure: 51.7% in control vs. 32.5% in intervention. This imbalance should be more acknowledged as a limitation in the discussion, and possibly explored through subgroup or sensitivity analyses even if you controlled by this variables in the models.

Pages 7 - 13, Results and then discussion: The drop in some outcomes at the 9-month follow-up seems important. This could suggest that while the program had promising short-term effects, those effects may not have been fully sustained. It would be useful if the discussion addressed this more. Not just in terms of outcome fluctuations, but as a reflection of how well the intervention was able to take root in the school context. These kinds of shifts over time often point to the limits of an intervention’s systemic integration, especially in complex and vulnerable environments. It also raises questions about whether additional support, reinforcement, or community involvement would be needed to keep those positive changes going. You are talking about effectiveness, so this is important to analyze in terms of the sustainability of the intervention over time.

Page 19, line 19 and in general: It would be more accurate to refer to the findings in terms of efficacy rather than effectiveness. While efficacy refers to the capacity of an intervention to produce expected changes under specific, often controlled conditions, effectiveness involves the extent to which those changes are sustained over time and contribute meaningfully to the intended purpose of the intervention in a real-world setting. Effectiveness imply that the intervention generates outcomes that are stable, lasting, and integrated into the system where it operates. Since the study evaluates short term changes in mental health indicators shortly after program implementation, and some of those changes appear to diminish by the 9-month follow-up, the findings are more appropriately interpreted as evidence of efficacy.

---

## [Reviewer Report]

Thank you for the opportunity to review the present study. Overall, the present study can make a significant and impactful contribution to the literature. Below are specific comments to strengthen the manuscript:

Abstract: Please provide a structured abstract

The abstract is missing some background information. Please add some context.

Introduction

The introduction heavily focuses on mental health symptoms among Afro-Colombian populations impacted by armed conflict. However, these are secondary intervention outcomes. The intervention is really focused on targeting “resilience, compassion, and prosocial behaviors”; therefore, the introduction should focus on those, instead of mental health symptoms.

Were cognitive impairments measured by trained staff members or self-disclosed?

Please expand the discussion of theoretical sampling and the reasoning for conducting focus groups over individual interviews.

Methods

Can you please provide more information about the intervention? For example, is it delivered by teachers? How are facilitators trained? By whom? How is competency of facilitators measured? Is the intervention delivered during school-hours or as an extracurricular activity? Does the intervention use evidence-based principles (e.g., Cognitive-behavioral therapy, interpersonal therapy? Mindfulness?). How long does each session last? How was the intervention developed? Has the intervention been pilot-tested?

More information about the data collection procedure is warranted. For instance, how were participants enrolled in the study? Did parents provide consent? Did adolescents provide ascent? Which institutional review board approved the study? Were surveys collected one on one with students in a private classroom of the school? How many trained psychologists collected the data? Who supervised them? How were they trained in data collection methods and in human research?

Please specify the fixed and the random effects in the models.

It is not clear how exposure to armed conflict was collected. Please provide this information.

More information about the qualitative data collection procedure and analysis is needed. For instance, how were participants selected to be part of the focus groups? Where were the focus groups conducted? When were the focus groups conducted? Were they conducted at baseline, endline and follow up? Were the participants the same? Who conducted them? What training did the person collecting the focus groups have? Had they worked with Afro-Colombian populations in the past? What did the semi-structured interview guide ask? Are there sample questions? Had they worked with conflict-affected youth? How long were the focus groups (i.e., range and median)? Did the participants receive anything for participating in data collection?

In terms of qualitative data analysis, it is necessary to provide a detailed account of the thematic analysis and provide a step-by-step description. Was a codebook developed? If so, how? Who led the analysis of the qualitative data? What training did the person leading the qualitative data analysis have? What about the people analyzing the qualitative data? Were the people analyzing the data involved in data collection as well? What training did they have? Were the focus groups recorded? If so, did participants consent to have the focus groups recorded? How many people were involved in the data analysis process? What methods were used to ensure trustworthiness?

Please describe the mixed methodology used in the study and how it was achieved (i.e., data triangulation was conducted in the results phase).

Results

Were the results similar with non-imputed results? Please provide non-imputed results as supplemental material.

In Table 1, to protect the confidentiality of participants, please do not use the name of the schools. Authors are encouraged to use School 1, School 2, etc.

A p-value of 0.000 is not feasible. Please revise this to reflect a p-value of <0.001, just like it is shown in Table 4.

Tables 2-4 say that “linear regression models” were conducted. However, the methodology indicates that linear mixed-effects models were used. Please clarify.

Regarding prosocial behavior, the authors indicate that “At baseline, both the intervention and control groups demonstrated similar prosocial behavior (PSB) scores, with a mean of approximately 85” and that “At 6 months postintervention, the intervention group’s scores rose significantly to 95.17 (SD = 7.40) and remained above baseline 9 months postintervention 81.42 (SD 6.33).” However, a mean score of 81 is lower than that of baseline (i.e., 85); therefore, the assertion that “remained above baseline 9 months postintervention” is not accurate. Please revise.

It is not clear why symptoms of anxiety and PTS were not included in the analysis as well. Authors are encouraged to conduct an analysis with anxiety and PTS and to determine whether the intervention also had statistically significant improvements in these symptoms.

The description of the Figure 4 does not fit the models represented in Figure 4. The descriptions of the measures and the numbers are completely off. Please review the name and the numbers of the figures on the Y-axis and revise the description of the figure and ensure they match.

Regarding the qualitative analysis, it is not clear what is being measured in terms of thematic saturation. This process needs to be explained in the methodology section.

The qualitative results do not share any insight as to how common these themes were in the overall group. Are the quotes representing something expressed by the majority of participants or just by a few? It is important to identify to what extent the majority of participants are identifying with an increased sense of self-compassion and empathy, rather than just a few.

Mixed method outcomes

The WDS is a screening tool, rather than a diagnostic tool. Please use language that reflects symptomology rather than diagnosis (e.g., depressive symptoms, rather than depression/positive cases of depression).

The triangulation results are promising; however, making inferences on the intervention’s lasting impacts mostly based on qualitative results (and keeping in mind that the quantitative results do not support these statements) is an overstatement. Please use more cautionary language, such as the qualitative results suggest the intervention can have lasting impacts.

Discussion

Try to make the implications of the intervention broader and not just for the Colombian population What about relating the promising results to the 2030 Healthy goals or more global mental health goals?

What other mental health interventions for youth have been used for Colombia? There are several other initiatives that are being tested in Colombia but are not mentioned in the Introduction or the Discussion section.

Minor comments:

Please use language that reflects symptomology rather than diagnosis (e.g., anxiety symptoms rather than anxiety; depressive symptoms, rather than depression; posttraumatic stress rather than posttraumatic stress disorder).

Please review the manuscript for punctuation errors and entire sentence repetition and typos. For example, page 3, line 33, there is a period, followed by a comma.

---

## [Reviewer Report]

Thank you for your careful revisions. The manuscript is significantly improved, with clearer methods, better integration of mixed-methods findings, and a more balanced discussion. Most prior concerns have been addressed. A few minor points remain that would strengthen the paper further:

Abstract

The abstract still contains redundant phrases (e.g., “evaluates examines the short-term efficacy of effectively…”). Please tighten for clarity and brevity.

The depression results should be stated more directly: emphasize the initial improvement followed by a rebound at 9 months, rather than presenting it as a minor observation.

Language and Style

Several sentences remain wordy or repetitive. For example, “students reporting continued sustained use of stress management” can be simplified to “students reported sustained use of stress management.”

Proofread for minor typographical errors (e.g., “postinterventionQuantitative” without spacing; double punctuation in some references).

Discussion

The discussion emphasizes resilience and anxiety improvements but gives less attention to why compassion, prosociality, and depression effects diminished by 9 months. Please expand on possible explanations (e.g., developmental stage of adolescents, external stressors, need for ongoing reinforcement, teacher delivery limitations).

Consider elaborating on scalability challenges beyond Tumaco. For example: What would be required to adapt the 3C program to other conflict-affected or resource-limited settings?

With these refinements, the paper will be in very strong shape for publication.

---

## [Reviewer Report]

The revised manuscript demonstrates substantial improvement, particularly in addressing effect attenuation and scalability challenges. However, the interpretation oscillates between appropriate scientific caution and unwarranted optimism, creating internal inconsistencies that undermine the manuscript’s credibility. The authors acknowledge significant limitations yet proceed to make policy recommendations that their own data do not fully support. Three major interpretive issues require attention before this manuscript can be recommended for publication.

Major Concerns

1. Disconnect Between Findings and Policy Claims

The manuscript presents a 50% success rate (three of six outcomes sustained at nine months: resilience, anxiety, and PTSD) yet characterizes the intervention as providing “essential” evidence for policy integration and a “replicable framework” for mental health promotion. This represents a fundamental misalignment between evidence and conclusions. Most concerning is the depression outcome, which showed complete rebound by nine months (61.2% positive screening versus 54.2% at baseline), yet the Impact Statement claims the program “offers a sustainable solution to reduce anxiety and trauma.” The conclusion asserts the study “provides robust evidence that targeted mental health interventions can produce substantial improvements” when the data demonstrate meaningful improvements for only half the measured outcomes, with effect sizes declining from large to moderate over the follow-up period.

I recommend substantially tempering the policy implications throughout the manuscript. The evidence supports characterizing this as “proof-of-concept requiring optimization” rather than a “scalable model ready for policy integration.” The intervention demonstrates feasibility and short-term efficacy for selected outcomes, which is valuable, but positioning it as policy-ready risks premature dissemination of an intervention that clearly requires refinement. Consider reframing this as foundational work opening a research trajectory rather than providing definitive answers for immediate implementation.

2. Inadequate Treatment of Contradictory Evidence

The depression findings present a striking paradox that receives insufficient critical attention. Quantitatively, positive screenings dropped dramatically to 21.3% at six months but rebounded to 61.2% at nine months—higher than the 54.2% baseline rate. Yet qualitatively, students reported sustained hope: "I still have hard days, but I know things can improve. I have hope now, which I didn’t have before." The Discussion attributes the rebound exclusively to external stressors (job losses, violence, harvest failures) and program limitations (teacher delivery, absent booster sessions) without considering three equally plausible alternative explanations.

First, the Whooley Depression Screening is a two-item tool designed for initial detection, not longitudinal tracking. It is highly sensitive to transient mood states and contextual factors. The end-of-year timing coinciding with acknowledged stressors could inflate positive screens without reflecting true sustained depressive disorder. Second, the intervention explicitly taught emotional awareness and regulation. Students may have become more skilled at recognizing and reporting depressive symptoms, making the apparent increase reflect improved mental health literacy rather than symptom worsening—a phenomenon recognized in the psychological intervention literature as “awareness effects.” Third, given the baseline imbalance (70.7% of controls versus 54.2% of intervention participants screened positive), some regression to the mean is statistically expected.

The qualitative finding that students maintained hope and reported using learned skills despite worsening screening rates suggests the Whooley may be capturing something other than the underlying emotional regulation capacities the program aimed to build. I recommend adding a paragraph explicitly acknowledging these measurement limitations and alternative interpretations. The current Discussion presents program failure as the only explanation when measurement artifact may be equally or more responsible for the observed pattern.

A similar contradiction appears with compassion and prosociality. Quantitatively, ECOM scores declined from 67.47 to 61.32 (below baseline) and PSB scores from 95.17 to 81.42 (below baseline). Yet qualitatively, students consistently described sustained empathetic listening, peer support, and compassionate interactions. The Discussion privileges the qualitative data, concluding that "students still exhibited compassionate behaviors, which suggests that the 3C program’s teachings had a lasting effect on social dynamics." This interpretive choice requires justification.

Three explanations merit serious consideration. First, the ECOM scale was validated in Mexican populations and may not adequately capture compassion as understood and practiced in Afro-Colombian cultural contexts, where compassion may be expressed through community solidarity rather than the scale’s focus on animal welfare and individual suffering relief. The authors acknowledge this limitation but dismiss it too quickly by noting they “triangulated” with qualitative data. Triangulation reveals measurement problems; it does not solve them. Second, students may have exhibited response shift bias—after enhanced self-awareness training, their nine-month self-evaluations may have employed higher standards than baseline assessments, making stable behaviors appear as declining scores. Third, the dramatic six-month spike in prosociality may have reflected social desirability bias during active intervention, with nine-month scores returning to a more realistic baseline.

I recommend explicitly reconciling these contradictions rather than selectively emphasizing data that support program effectiveness. If qualitative and quantitative data diverge, both possibilities require equal consideration: either behaviors genuinely declined despite self-reports, or measurement validity problems make the quantitative data unreliable for this population. The current approach of highlighting qualitative data when it supports program success while emphasizing quantitative data for anxiety and resilience creates an impression of confirmatory bias.

3. Baseline Imbalances Dismissed Too Readily

The authors acknowledge baseline imbalances in gender (71.3% female in control versus 42.0% in intervention) and conflict exposure (51.7% versus 32.5%) but dismiss concerns by noting that “results adjusted for gender and formal interaction tests suggest minimal risk of bias.” This conflates two distinct statistical concepts: effect modification (do intervention effects differ by subgroup?) and confounding (did the imbalance bias effect estimates?). The interaction tests address the former but do not eliminate the latter.

The imbalances are consequential. Males showed significantly lower compassion (β = −1.9, p = 0.022) and prosociality (β = −1.09, p = 0.040) than females. The control group also started with worse mental health burden (70.7% versus 54.2% depression-positive at baseline). These patterns raise the possibility that some proportion of the observed intervention effects—particularly the large effects at six months—may reflect regression to the mean in a control group that started with higher risk profiles. Covariate adjustment in regression models reduces but cannot fully eliminate confounding from such substantial imbalances, especially when the imbalanced variables correlate with outcomes and baseline risk differs between groups.

I recommend revising the limitations section to acknowledge that cluster randomization at the school level, while necessary for intervention integrity, limited ability to achieve balanced groups, and that some observed effects may be partially confounded by these imbalances. The current treatment reads as defensive justification rather than transparent acknowledgment of an inherent limitation of cluster-randomized designs. Consider adding sensitivity analyses stratified by gender or using propensity score methods to assess robustness of findings to these imbalances.

Moderate Concerns

4. Generic Scalability Discussion Disconnected from Study-Specific Insights

The added paragraphs on scalability (lines 1885-1917) appropriately cite relevant literature and identify important implementation challenges including cultural adaptation, teacher training infrastructure, parental involvement, and integration with clinical services. However, these paragraphs read as a generic discussion of scaling mental health interventions in low- and middle-income countries rather than being grounded in this study’s specific context and findings. The Discussion would be substantially strengthened by leading with insights derived directly from your data.

Your study achieved 95% retention at both follow-up points—exceptional for conflict-affected populations. What made this possible? School integration? Community partnership? Compensation? Teacher credibility? These success factors merit discussion as they represent critical insights for scalability that generic citations cannot provide. Similarly, you observed gender-differentiated engagement, with predominantly female groups showing higher compassion and prosociality. This suggests programs may need tailored activities to engage male adolescents effectively—a study-specific insight more valuable than general statements that “implementation research is needed.”

Your data also revealed that teacher delivery was insufficient for depression (complete rebound) but effective for anxiety and PTSD. This has direct implications: scalable models should embed school programs within stepped-care systems that link students screening positive for depression to mental health specialists rather than relying solely on teacher-delivered universal prevention. Additionally, your nine-month follow-up timing revealed vulnerability to seasonal stressors (year-end job losses, harvest cycles, violence escalation). This suggests booster sessions should be scheduled before predictable high-stress periods—a concrete, actionable recommendation grounded in your findings rather than theoretical speculation about what “may be more feasible” in other settings.

I recommend substantially revising the scalability section to prioritize study-specific insights while using literature to contextualize rather than lead the discussion. This would transform the section from theoretical speculation to evidence-based guidance.

5. Insufficient Exploration of Mechanisms

The Discussion describes what happened but insufficiently explores why. Your intervention had three stated components: intrapersonal (“with me”), interpersonal (“with you”), and community (“with everyone”). The pattern of results illuminates which mechanisms sustained beyond active intervention. Resilience gains persisted at nine months, suggesting intrapersonal regulation skills became internalized and self-reinforcing—students could practice breathing exercises, cognitive reframing, and stress management independently. In contrast, compassion and prosociality declined, suggesting interpersonal and community skills require continuous social reinforcement and practice opportunities that dissipated post-intervention. This pattern aligns with behavioral maintenance theory: skills requiring only individual practice sustain better than those dependent on reciprocal social interactions.

Similarly, the gender differential warrants mechanistic exploration. Why did males benefit less? Did boys engage less in activities? Are socioemotional curricula inherently more compatible with feminine socialization in this cultural context? Does the curriculum content resonate less with masculine identity construction in conflict-affected communities? These are not merely speculative questions—your qualitative data could address them through systematic analysis of engagement patterns and thematic differences by gender.

I recommend adding a mechanisms subsection that uses your pattern of findings to generate testable hypotheses about active ingredients. This would strengthen the scientific contribution by moving beyond demonstrating that the intervention worked for some outcomes to explaining why it worked differentially across outcomes and populations.

6. Underutilization of Mixed-Methods Design

The manuscript presents qualitative findings primarily as supportive illustrations of quantitative results rather than as co-equal evidence requiring integration and reconciliation. Throughout the Discussion, the pattern is: (1) state quantitative result, (2) provide supportive qualitative quote, (3) move on. For example: “This trend aligns with qualitative reflections in which students described the lasting impact... One student reported, ‘Even months later, I still use the breathing exercises when I get nervous, and they help me stay calm.’” This is confirmatory rather than analytical.

Your mixed-methods design enables deeper integration. Did students who reported sustained strategy use in qualitative interviews show better quantitative outcomes? If you coded qualitative responses for resilience themes and tested whether theme presence correlates with CD-RISC scores, this would validate both data types and identify which specific strategies drove effects. Similarly, divergent cases illuminate mechanisms: Who improved quantitatively but did not report skills qualitatively? Who maintained skills qualitatively but declined quantitatively? These cases reveal measurement issues or unmeasured mechanisms that could inform program refinement.

The qualitative data could also address why some students sustained gains while others declined. Among the intervention group, what distinguished students who maintained benefits from those who returned to baseline? The qualitative interviews likely captured differences in peer support, family reinforcement, personal motivation, or continued practice that explain outcome heterogeneity. These insights have direct practical implications for identifying students who need additional support.

I recommend deeper analytical integration of the two data streams. This would transform the qualitative component from decorative to essential and would substantially strengthen the mixed-methods contribution of the manuscript.

Minor Issues

The manuscript would benefit from reorganizing the Discussion to follow a more logical progression: (1) principal findings (factual summary), (2) mechanisms explaining differential patterns, (3) measurement validity considerations, (4) methodological limitations including baseline imbalances, (5) implications for program refinement, (6) contextualization of findings, and (7) future directions following a clear optimization-to-effectiveness-to-scaling sequence. This structure would prevent the current oscillation between optimism and caution and create a more coherent narrative arc.

Additionally, some statements require softening. The phrase “robust evidence” (line 1920) overstates findings given measurement validity concerns. “Substantial improvements” should be qualified to note they occurred for only half of outcomes and effect sizes declined over time. The Impact Statement’s claim of “sustainable solution” contradicts the acknowledged need for booster sessions and the depression rebound.

Recommendation

The authors have conducted rigorous research on an important topic and the manuscript has improved substantially through revision. However, the interpretation requires recalibration to match the evidence. The data support characterizing this as a promising intervention requiring optimization—not a scalable model ready for policy integration. The evidence demonstrates feasibility and short-term efficacy for selected outcomes in one Afro-Colombian community, which represents valuable proof-of-concept. Positioning this work as foundational research opening a trajectory for iterative program development would strengthen rather than weaken the scientific contribution. It would also prevent potential harm from premature dissemination of an intervention that clearly requires refinement, particularly regarding depression outcomes, sustainability beyond nine months, and engagement of male adolescents.

I recommend major revisions addressing: (1) tempering policy claims throughout to match evidence, (2) adding nuanced interpretation of depression findings acknowledging measurement limitations, (3) reconciling quantitative-qualitative contradictions rather than selectively emphasizing supportive data, (4) transparently acknowledging that baseline imbalances may partially confound effects, (5) grounding scalability discussion in study-specific insights, and (6) deepening mechanistic exploration and mixed-methods integration. With these revisions, this manuscript would make an important contribution to the literature on school-based mental health promotion in conflict-affected settings.

---

## [Editor Report]

The authors substantially revised the manuscript, adequately addressing all (major, moderate and minor) concerns.